# The Applications and Pitfalls of Cone-Beam Computed Tomography-Based Synthetic Computed Tomography for Adaptive Evaluation in Pencil-Beam Scanning Proton Therapy

**DOI:** 10.3390/cancers15205101

**Published:** 2023-10-22

**Authors:** Pingfang Tsai, Yu-Lun Tseng, Brian Shen, Christopher Ackerman, Huifang A. Zhai, Francis Yu, Charles B. Simone, J. Isabelle Choi, Nancy Y. Lee, Rafi Kabarriti, Stanislav Lazarev, Casey L. Johnson, Jiayi Liu, Chin-Cheng Chen, Haibo Lin

**Affiliations:** 1New York Proton Center, New York, NY 10035, USA; ptsai@nyproton.com (P.T.); bshen@nyproton.com (B.S.); azhai@nyproton.com (H.A.Z.); francisyu@nyproton.com (F.Y.); csimone@nyproton.com (C.B.S.II); ichoi@nyproton.com (J.I.C.); cjohnson@nyproton.com (C.L.J.); liujiayi_77@icloud.com (J.L.); chen.ccc@gmail.com (C.-C.C.); 2Proton Center, Taipei Medical University, Taipei 11031, Taiwan; 214083@h.tmu.edu.tw; 3Department of Radiation Oncology, Taipei Medical University, Taipei 11031, Taiwan; 4California Protons Center Therapy Center, San Diego, CA 92121, USA; christopher.ackerman@californiaprotons.com; 5Department of Radiation Oncology, Memorial Sloan Kettering Cancer Center, New York, NY 10065, USA; leen2@mskcc.org; 6Department of Radiation Oncology, Montefiore Medical Center, Bronx, NY 10467, USA; rkabarri@montefiore.org; 7Department of Radiation Oncology, Icahn School of Medicine at Mount Sinai, New York, NY 10029, USA; stanislav.lazarev@mountsinai.org

**Keywords:** synthetic CT, proton therapy, cone-beam computed tomography, pencil-beam scanning, adaptive

## Abstract

**Simple Summary:**

For proton therapy, verification CT (vCT) scans are routinely acquired to ensure the accuracy and precision of treatment delivery, as this approach allows clinicians to monitor and adapt treatment plans based on potential changes to the patient’s anatomy and tumor. This study investigates synthetic CT’s (sCT) potential as an alternative to vCT scans, focusing on its reliability across various treatment sites. Synthetic CT could offer a more efficient approach and possibly enhance patient experience by reducing the necessity for frequent vCT scans. The consistency between sCT and vCT in terms of image quality and dosimetric impact is crucial. It will allow clinicians to monitor and adjust treatment plans promptly and more accurately based on patients’ anatomical changes, potentially optimizing treatment processes and outcomes. The insights from this study are pivotal for refining clinical practices and fostering advancements in treatment strategies, ultimately aiming at precision and accuracy in adaptive treatment.

**Abstract:**

Purpose: The study evaluates the efficacy of cone-beam computed tomography (CBCT)-based synthetic CTs (sCT) as a potential alternative to verification CT (vCT) for enhanced treatment monitoring and early adaptation in proton therapy. Methods: Seven common treatment sites were studied. Two sets of sCT per case were generated: direct-deformed (DD) sCT and image-correction (IC) sCT. The image qualities and dosimetric impact of the sCT were compared to the same-day vCT. Results: The sCT agreed with vCT in regions of homogeneous tissues such as the brain and breast; however, notable discrepancies were observed in the thorax and abdomen. The sCT outliers existed for DD sCT when there was an anatomy change and for IC sCT in low-density regions. The target coverage exhibited less than a 5% variance in most DD and IC sCT cases when compared to vCT. The D_max_ of serial organ-at-risk (OAR) in sCT plans shows greater deviation from vCT than small-volume dose metrics (D0.1cc). The parallel OAR volumetric and mean doses remained consistent, with average deviations below 1.5%. Conclusion: The use of sCT enables precise treatment and prompt early adaptation for proton therapy. The quality assurance of sCT is mandatory in the early stage of clinical implementation.

## 1. Introduction

In proton therapy, the precision of patient setup and consistent anatomy throughout the treatment course are paramount to achieving optimal therapeutic outcomes. Proton therapy is favored for its ability to maximize organ-at-risk (OAR) sparing due to the absence of an exit dose. However, any inter-fraction anatomical changes could significantly compromise target coverage or increase the dose to OARs. Hence, verification CT (vCT) scans are routinely acquired to assess dosimetric accuracy for target and OARs, and they serve as the most recent anatomy reference for initiating adaptive planning. As image-guided radiation therapy (IGRT) [1] gains prominence, tools like cone-beam computed tomography (CBCT) [2] have become essential for the daily setup and monitoring of anatomical variations. Yet, the limitations of CBCT, including significant scatter artifacts [3,4,5,6], beam hardening [7], and suboptimal image quality [8], constrain its utility for accurate proton stopping power estimation. To overcome this, synthetic CT (sCT) generated from CBCT has been developed to assess anatomical changes and support adaptive treatment planning. Using sCT can potentially reduce the demand for vCT scans during treatment, optimizing clinic efficiency and the adaptive evaluation process. Given the growing interest in this field, understanding the potential and limitations of these techniques in routine clinical workflows remains a priority.

Research has consistently highlighted the potential of sCT in photon-based radiation therapy, showcasing its adaptability across various areas, including the head and neck (H&N) [9,10], breast [11], thorax [12], and pelvis [12,13] regions. Yet, despite these advancements, the integration of synthetic CT into proton therapy—a domain marked by its unique sensitivities—remains circumscribed to limited data across a select few treatment sites. Several literatures [14,15,16,17,18,19] demonstrate that CBCT-based sCT generation methods generally fall into three categories: direct-deformable (DD), image-correction (IC), and artificial intelligence (AI). Veiga et al. [15,16] explored the DD method with a focus on adaptive proton therapy for lung cancer patients. Concurrently, Kurz et al. [17] investigated the IC approach, emphasizing its potential for CBCT-based dose calculation in H&N cancer patients. Reiners et al. [18] explored both the DD and IC methods in the RayStation Treatment Planning System (TPS) (version 12A), assessing their utility for daily dose monitoring and triggering plan reviews for H&N cancer. They concluded that synthetic CT is a more efficient and accurate adaptive workflow in proton therapy. In addition, several publications [19,20,21,22,23,24,25] ventured into AI-based synthetic CT generation that is specifically geared toward proton therapy application. Landry et al. [19] compared VMAT (Volumetric-Modulated Arc Therapy) and IMPT (Intensity-Modulated Proton Therapy) utilizing AI method (U-Net). Their results showed varying outcomes based on the training set choice, with only one approach proving suitable for proton therapy over photon radiation therapy. Spadea et al. [26] evaluated the merits of deep convolutional neural networks (DCNN) against traditional methods like deformable image registration (DIR) and analytical image-based correction (AIC) using the RayStation TPS (v7.99). Several studies have shown that both DD and AI methods demonstrate the capacity to produce sCT of excellent image quality [13], less noise and artifacts [27], and excellent dosimetric accuracy [28].

While strides have been made in sCT research, challenges persist in practical clinical settings. Taasti et al. [29] investigated both the DD and IC methods, specifically in the thoracic region. Their findings showed that false negatives could occur, and indications for adaptation might be triggered solely by one of the image modalities. This brings to light the possibility of discrepancies and controversial outcomes in clinical scenarios. While many authors and institutions have introduced AI methods, users from different institutions often find it challenging to validate the sCT generation processes governed by neural networks. Verifying the compatibility of model-generated sCT images with original training datasets, employed hyperparameters, and variations in deep learning network design remains complex. Recognizing these limitations, Oria et al. [30] proposed a quality control solution to identify the potential failure cases or outliers in the context of AI sCT, enabling users to swiftly identify inconsistencies or outliers, especially within the context of H&N cancer. While several such studies exist, most revolve around an in-house-developed model, a single system, or a research version of commercial systems. Moreover, their focus predominantly remains confined to individual treatment sites. Many studies have provided valuable insights during the developmental phases, yet the ongoing challenge remains in assessing the performance of commercially available sCT solutions in clinical settings.

This study aims to evaluate sCT derived from commercialized image information systems in a clinical environment. Centering on the role of CBCT-based sCT in pencil-beam scanning (PBS) proton therapy’s adaptive planning process, we scrutinize two commercial systems covering seven major treatment areas. We also aim to evaluate the effectiveness of sCT and its potential challenges in a clinical setting.

## 2. Materials and Methods

In this study, we selected patients who underwent PBS proton therapy across various treatment sites, including H&N, brain, lung, breast, pelvis, high-risk prostate with pelvic lymph node irradiation (Prostate + LN), and abdomen. For treatment planning design, the typical beam arrangement consisted of two to three beam angles, tailored to the specific tumor locations and their proximity to OARs. The beam angle selection follows the short, homogeneous, and stable beam path rule, and has low sensitivity to respiratory motion and setup uncertainties. In the instance of complex H&N cases where the disease extended into the nasal sinus region, it became necessary to utilize more than the standard three beams. In certain cases, a “no-fly zone” or beam-specific target region strategy was used when appropriate. A relative biological effectiveness (RBE) of 1.1 was applied in the TPS for the proton treatment plan, and all doses reported as ‘Gy’ were understood to represent ‘Gy (RBE)’. The initial criterion for patient inclusion was based on matching CBCT and vCT scans acquired on the same day. These patients were then randomly selected from our database as potential candidates. The planning CT (pCT) and vCT images were obtained using a Siemens Somatom CT simulator (Siemens, Forchheim, Germany), while the CBCT images were captured in the patient’s treatment position using a Varian ProBeam gantry-mounted imager (Varian Medical Systems, Inc., Palo Alto, CA, USA).

The CBCT images, vCT, and pCT (including structure set) of the patient, along with online match rigid registration, were transferred to two different commercialized image information systems: Velocity 4.1 (Varian Medical Systems, Palo Alto, CA, USA) and MIM Maestro 7.1.4 (MIM software, Cleveland, OH, USA). The sCTs were then generated utilizing the automatic workflow provided by the image information systems.

### 2.1. Direct-Deformable Synthetic CT

The Velocity image information system utilizes the DD method. In this study, the built-in assessment workflow in Velocity using the navigator “Single Plan Generation: ACTOR” was applied. The sCT generation procedure begins with integration with a pCT and related structures, alongside at least one CBCT scan taken during treatment. A rigid online match registration is subsequently applied between each CBCT scan and the pCT. A deformable registration is then constructed between the pCT and CBCT.

A region-of-interest (ROI) with the entirety of the CBCT’s FOV was used to ensure that the pCT anatomy was fully deformed via a multi-pass corrected deformable image registration (DIR) algorithm. Following this, the sCT is produced by merging the DIR-adjusted pCT inside the CBCT field-of-view (FOV) and stitching the pCT outside the FOV. A DD sCT, called adaptive CT (aCT) in the Velocity system, is generated by reshaping the pCT based on each treatment volume’s specific deformable. Upon the synthesis of the DD sCT images, the workflow automatically pairs the relevant treatment plans and structures to the DD sCT.

### 2.2. Image-Correction Synthetic CT

In the MIM image information system, the IC method [31,32] is employed for sCT generation. Instead of distorting the CBCT, it enhances its corrections, named “enhancement CBCT”. After transferring the pCT, CBCT, and online match registration to the MIMpacs workspace for sCT generation, the presence of the online registration between the CBCT and pCT triggers the IC sCT synthesis via the MIM assistant workflow automatically. Once the process is completed, the generated IC sCT becomes accessible in the MIMpacs workspace.

The workflow’s initial step addresses the shading artifact, a common characteristic affecting CBCT image quality. Simultaneously, the voxel values and the intensity of CBCT are fine-tuned to match the Hounsfield unit (HU) values of the pCT, ensuring their suitability for dose calculation. This adjustment facilitates a precise voxel-to-voxel correlation via the multi-modality (CT-CBCT) deformation algorithm. Concluding the process, a deformable merge combines the refined CBCT with the pCT. In this phase, the refined CBCT is preserved as-is, while the deformation from the pCT expands from this refined CBCT to the outer limits of the pCT’s FOV, finalizing the creation of the IC sCT.

### 2.3. Synthetic CT Evaluation

After the sCT was generated, it was transferred to the Eclipse TPS (Varian Medical Systems, Palo Alto, CA, USA) along with the associated structure contours and treatment plan. Subsequently, the sCT was assigned the same CT calibration curve as its corresponding vCT to ensure consistency. Forward dose calculation was performed on the sCT with pCT treatment plan parameters using the proton convolution superposition (PCS) algorithm version 15.6 in the Eclipse TPS.

The sCTs were evaluated on both dosimetric impact and image quality. A visual inspection was first conducted on each sCT paired with its corresponding vCT/CBCT image to discern any discrepancies. If a sCT image either failed to be generated or displayed significant anomalies that influenced the vicinity of the treatment region, it was categorized as an outlier and excluded from subsequent dosimetric studies. For every treatment site, at least ten cases were considered for this dosimetric assessment. After the doses were re-calculated on the DD sCT and IC sCT, several target and OAR dose constraints were extracted and compared against the reference vCT plan. Among the clinical metrics, D0.1cc denoted the dose received by a minimum of 0.1 cc of serial OARs, such as the spinal cord or brainstem. The mean or volumetric doses were computed for parallel OARs, such as the lung or kidney. The dose difference between the sCT plan and its paired vCT plan was also assessed, with the results presented as the mean with standard deviation. A paired *t*-test was performed between the datasets of the two sCT types, with a significance level set at *p* < 0.05. If the dosimetric results of the sCT plan were not comparable to the same day’s reference vCT plan, these cases were identified for further investigation. The possible underlying causes and limitations of sCT will be elaborated upon in the results and discussion of these cases.

For the assessment of image quality, a comprehensive evaluation of sCT and vCT images was carried out using quantitative metrics: mean squared error (MSE), peak signal-to-noise ratio (PSNR), and structural similarity index measure (SSIM). MSE was employed to measure the average squared difference between the pixel values in the sCT and vCT images, providing a quantitative assessment of image dissimilarity.

The MSE equation is defined as follows:(1)$$MSE=\frac{1}{N}\sum_{i,j}{\left(S\left[i,j\right]-V\left[i,j\right]\right)}^{2}$$
where (i,j) denotes the pixel coordinates, N is the total number of pixels in the image, V[i,j] presents the pixel value in the vCT image at coordinates (i,j), and S[i,j] represents the pixel value in the sCT image at coordinates (i,j).

PSNR was utilized to assess the quality of the sCT reconstruction, considering the pixel value differences and the dynamic range of the images. Higher PSNR values indicate superior image fidelity.

The PSNR equation is as follows:(2)$$PSNR=10\log_{10}\left(\frac{{MAX}^{2}}{MSE}\right)$$
where MAX is the maximum possible pixel value (3070 for the 12-bit CT image in this study) for the given data type, and MSE is the mean squared error between the sCT and the vCT images.

Furthermore, SSIM was applied to consider luminance, contrast masking, and structural information, offering insights into the perceptual similarity between the sCT and vCT images. The simplified SSIM equation is expressed as:(3)$$SSIM={\left(L(V,S)\times C\left(V,S\right)\times S\left(V,S\right)\right)}^{\alpha }$$

In this equation, L(V, S), C(V, S), and S(V, S) represent the comparisons of luminance, contrast, and structure, respectively. The parameter α adjusts the relative importance of these components. For a more detailed explanation of SSIM, refer to the work by Kida et al. [13]. A higher SSIM score signifies a closer resemblance between the two image types regarding pixel values and visual features. These metrics offer a comprehensive assessment of sCT image quality, which will be evaluated and categorized by the treatment site.

## 3. Results

From the cohort, CT images from 81 patients underwent image quality assessment for two sets of sCT per patient. Of those, a total of 70 patients were included in the group analysis to allow for 10 cases at each anatomy site. Eleven cases were identified as outliers, with three instances experiencing failures in the sCT generation workflow due to the presence of severe metal artifacts. Specifically, one case in the pelvis group was affected by bilateral hip prostheses, while two cases in the breast group were impacted by a breath-holding device and the tissue expander. These artifacts arising from the high-density artificial materials on the CT images were the primary factors for the workflow challenges. Additionally, two outlier cases were identified where the automatic workflow failed in the proper prorogations of the pCT contour to sCT. Furthermore, there were outliers associated with image-related issues, including isocenter position shift errors and unaccounted-for anatomical changes in the sCT.

### 3.1. Head and Neck

Dose metrics such as D_95%_, D_0.1cc_, and D_mean_ were used to evaluate the dose difference in various structures such as CTV, brain, brainstem, chiasm, oral cavity, left/right parotid, and spinal cord for the H&N tumor site. The means and deviations of the dose differences between sCT and vCT are shown in Figure 1. CTV D_95%_ showed lower variation for DD sCT (0.14 ± 2.94%) compared to the IC sCT (−2.71 ± 5.73%) in the H&N region. The IC sCT showed an HU value discrepancy (~200 HU on average) in the low-density region (e.g., air cavity and head cushion) in two patients, which resulted in the CTV D_95%_ having more than a 10% dose difference when compared to the vCT plan. The *p*-value showed no statistically significant difference between DD and IC sCT.

### 3.2. Brain

For the brain group, a variety of dose metrics for CTV, brain, brainstem, spinal cord, optic structures, and temporal lobes are presented in Figure 2. Remarkably, the D_95%_ CTV target was a less than 1% deviation for DD sCT (0.31 ± 0.75%) and IC sCT (−0.25 ± 1.11%), showing good agreement compared to the vCT plan. For the optic OARs, both DD and IC sCT showed similar deviation and dose metric values without statistical significance. However, one case showed a more than 2 Gy variation in D0.1cc within the chiasm, brainstem, and optic structure compared to the vCT. This resulted from tissue discrepancies on DD sCT and HU value discrepancies (~200 HU on average) on IC sCT compared with the vCT. Overall, the *p*-value assessments indicated no significant statistical differences between the DD and IC sCT. The deviations for both targets and OARs, when compared to the vCT plan, were minimal. The left temporal lobe was the only OAR demonstrating statistical significance (*p* = 0.03). However, this variation is considered negligible with a dose level difference of less than 0.5 Gy.

### 3.3. Lung

For the lung group, the dose metrics were evaluated for various anatomical structures, including the CTV, total lung minus GTV, heart, spinal cord, and esophagus. As shown in Figure 3, large deviations were observed for the smaller volumetric evaluator, D_0.1cc_, especially in the heart (−2.17 ± 6.46 Gy for DD sCT and −6.90 ± 13.29 Gy for IC sCT) and esophagus (−0.97 ± 4.81 Gy for DD sCT and −2.85 ± 7.01 Gy for IC sCT). However, when evaluating these OAR structures using the mean dose, the deviations from the vCT remained within acceptable limits. Specifically, the D_mean_ for the heart was −0.07 ± 0.55 Gy for DD sCT and 0.25 ± 0.62 Gy for IC sCT, and for the esophagus, this was −0.15 ± 0.59 Gy for DD sCT and −0.35 ± 0.69 Gy for IC sCT. Most dose metrics between DD and IC sCT showed no significant difference. A significant discrepancy in CTV coverage was observed between the DD sCT and IC sCT plans when compared to the vCT plan (*p* = 0.01). A similar trend was also evident in the total lung GTV metrics, specifically for mean dose, V_5Gy_, V_10Gy_, and V_20Gy_ (*p*-value < 0.01).

### 3.4. Breast

For the breast group, Figure 4 illustrates the evaluation of dose metrics for various contour structures, including CTV, esophagus, total lung, and heart. Overall, both sCT plans demonstrated minimal deviation from the vCT plan in terms of target coverage. However, the DD sCT displayed a wider spread (standard deviation) than the IC sCT concerning the CTV target coverage. In line with the findings from the lung group, the D_0.1cc_ indicated more significant deviations in the IC sCT for the esophagus (−0.98 ± 3.99 Gy for DD sCT vs. −2.26 ± 3.36 Gy for IC sCT). Regarding OARs, there were no statistically significant differences between DD sCT and IC sCT when compared with vCT.

### 3.5. Pelvis

For pelvic anatomical sites, CTV, bladder, rectum, bowel bag, and cauda equina were included in the evaluation. As shown in Figure 5, minimal dose differences were observed for the volumetric dose evaluator of both right and left femoral heads. This minor discrepancy arose from the near-zero dose values of these metrics across all vCT and both sCTs, rather than indicating a precise dose representation by the sCTs when benchmarked against the vCT results. The majority of cases exhibited a difference of less than 1% in D_95%_ CTV coverage in relation to the vCT plan. However, two cases showed a CTV D_95%_ deviation exceeding 5%. These anomalies were attributed to significant artifacts present in the bowel structure on the original CBCT, which, being in proximity to the CTV target at the proton beam’s distal end, influenced the dosimetric outcome of the target coverage. Although the bladder’s Dmean was small, it displayed significant fluctuations, a phenomenon attributed to bladder volume changes between the vCT and CBCT scans. In the case of the rectum, the V_55Gy_ for both sCT variants exhibited substantial variation (−2.48 ± 7.43% for DD sCT and −5.46 ± 8.80% for IC sCT). The *p*-value analysis of the dose metrics revealed no significant differences between DD sCT and IC sCT plans when compared with the vCT plan.

### 3.6. Prostate with Pelvic Lymph Node Involved

For the prostate + LN group, various anatomical structures, such as CTV_High (prostate), CTV_Low (pelvic wall lymph node), bladder, rectum, and femur heads, were included in the analysis. The mean and deviation values for each dose evaluator are plotted in Figure 6.

All deviations for the CTV D_95%_ values in the sCT plans stayed within a 2% range in relation to the vCT plan. A noteworthy deviation is observed for the V_32.8Gy_ of the bladder (3.13 ± 7.36%) and V_14Gy_ of the rectum (1.79 ± 10.21%) in the DD sCT plan. Statistical assessments revealed no discernible differences between DD and IC sCT. When comparing the sCT and vCT plans for OAR, the deviations were consistently slight, with the median deviation approximating 0%.

### 3.7. Abdomen

For the abdomen group, we evaluated the dosimetric results for several contour structures, including the CTV, kidneys, liver, bowels, and stomach. These results from the abdomen group are depicted in Figure 7. The bowel structure’s evaluation considered areas proximate to the target volume, such as the small bowel, large bowel, or bowel bag. In three out of ten cases, when the bowel structure was situated at the beam’s distal end and directly adjacent to the target, there was substantial variation in the volumetric dose difference for V_15Gy_. This ranged from −13.86 to 13.72 cc for the DD sCT plan and −19.00 to 26.28 cc for the IC sCT plan. Additionally, three cases displayed deviations exceeding 10% for the bowel when comparing the IC sCT to the vCT plan. The t-test indicated no statistically significant difference between DD and IC sCT.

### 3.8. Image Quality

From the MAE analysis across all treatment sites, only the lung sCT demonstrated a statistically significant difference between DD sCT (18.70 ± 4.70) and IC sCT (28.10 ± 7.60) with a *p*-value of < 0.001. This indicates that, according to the MAE analysis on a pixel-by-pixel basis, DD sCT offers more superior image consistency than IC sCT when compared to vCT. In terms of PSNR, which quantifies image fidelity, DD sCT consistently outperforms IC sCT across all treatment sites. Notably, several treatment sites exhibited statistically significant differences in PSNR between DD sCT and IC sCT: H&N (*p* < 0.001), brain (*p* = 0.01), lung (*p* < 0.001), and breast (*p* < 0.001). These findings underscore the superior image quality offered by DD sCT in these specific anatomical regions. In evaluating SSIM, both DD sCT (0.85 ± 0.13) and IC sCT (0.86 ± 0.13) displayed consistent image fidelity and structural similarity, with no statistically significant differences across all treatment sites.

A visual inspection revealed localized discrepancies in certain areas (Figure 8) between the sCT images and the original patient anatomy. Although these instances are infrequent, they may potentially impact the dosimetry results. These discrepancies can be categorized into four primary groups: aliasing artifacts in synthetic CT, distortion of the support couch structure, inconsistencies in HU values within low-density regions, and inaccuracies in replicating the patient’s same-day anatomy.

In assessing synthetic CT image quality, aliasing is most commonly seen near the edge of the CBCT field of view, especially when it overlaps with the original planning CT. This is particularly evident in brain cases for both sCT types. For instance, Figure 8A shows aliasing in the anterior–posterior direction close to the CBCT’s edge, leading to a distorted brain shape towards its posterior. Similarly, the superior–inferior aliasing in Case 2 depicted in Figure 8B induces an isocenter position shift error, yielding skewed dosimetric outcomes. Figure 8C also reveals a substantial deformation of the support couch structure in the DD sCT.

For IC sCT, certain situations display substantial HU discrepancies between the vCT and the IC sCT, especially in low-density regions. The resulting synthetic CT often has an HU approximately 100–200 units higher than the HU value of vCT in areas such as the head cushion, nasal cavity air region, and vacuum bag region, as illustrated in Figure 9A–C. It is advised to rectify such HU discrepancies before advancing with calculations on the synthetic CT images.

For the DD sCT, under certain circumstances, the DD method does not accurately capture the intricate anatomical details present in the CBCT taken on the same day, especially near heterogeneous interfaces. These discrepancies are especially noticeable when a patient experiences substantial anatomical changes throughout the treatment course. Figure 10 highlights four instances: (A) tumor shrinkage in the nasal cavity; (B) expansion of the chest wall due to respiratory fluctuations; (C) shifts in bowel air content; and (D) streaking artifacts, where the discrepancies reach beyond the immediate area of the metal artifact, impacting the air cavity within the rectal region. These pronounced HU value variations arise from the stark contrast between tissues or materials exhibiting low and high HU values.

### 3.9. Summary

The standard deviation in OAR dose deviation can be up to 2 Gy between the plans calculated on the corresponding sCT and vCT. In terms of target coverage, the percentage dose evaluator typically demonstrates a deviation of 5% or less from the reference vCT plan, assuming no image discrepancies. The CTV D_95%_ is less than a 5% difference for 86% DD sCT cases and 89% IC sCT cases compared to the vCT results. For absolute dose evaluation, the average dose difference between the sCT plan and vCT plan hovers around 1–2 Gy. For the cases without observable image quality outliers, it is observed that for non-moving target anatomical sites such as the brain and H&N, the target dose evaluator on sCT achieved a deviation of less than 1.5% from vCT. In soft tissue anatomical sites like the breast, pelvis, and prostate + LN, the deviation typically hovered around 2% compared to vCT. However, in moving target anatomical sites like the lung and abdomen, the agreement with vCT was around 5%. When the image outlier impact area is in the proton beam path, the deviation can be easily greater than 5%, thus triggering further investigation.

Upon investigating the point (Dmax) and small-volume dose (D_0.1cc_) metrics across different OARs, it becomes evident that small-volume dose serves as a more suitable dose evaluator for guiding adaptive treatment decisions with sCT. For the serial OARs, such as the brainstem, spinal cord, and cauda equina, D_0.1cc_ consistently exhibited lower deviations compared to Dmax within the same type of sCT. Therefore, D0.01 cc is the preferred choice over Dmax as the crucial dose metric when examining serial OARs to prevent misleading dose deviation. In the moving target region, the volumetric and mean dose evaluator for parallel OARs (e.g., heart and esophagus) in the sCT plans is generally recommended where it is consistent with the vCT plans, with less than 1.5% deviations on average.

## 4. Discussion

The adaptation protocol using sCT generated from CBCT allows us to closely monitor and adapt treatment plans as needed. When employing sCT, we observe several potential improvements. First, potential patient positioning differences exist in vCT, acquired with re-setup in the simulation room. In contrast, CBCT captures the patient’s position on the treatment table, where patient positioning is fine-tuned through KV images or CBCT. Second, it streamlines plan evaluations, reducing the necessity for multiple vCT scans and minimizing additional radiation exposure to the patient. Third, it expedites the detection of anatomical changes, enhancing our ability to promptly adjust treatment plans. In summary, the integration of synthetic CT into our adaptation protocol holds promise for enhancing both the quality and timeliness of adaptive planning, ultimately leading to improved treatment outcomes for the patients.

### 4.1. Image Quality

The sCT discrepancies are often spotted at the image border between CBCT and pCT. Such image aliasing effects, particularly when the proton beam traverses through, can alter the proton ranges. For brain tumor patients, a comprehensive comparison of brain contour volume between CBCT and sCT is essential. Distortions in the support couch structure region of the DD sCT result from the DIR algorithm’s attempt to align the pCT and CBCT, which possess distinct couch bases. This can be addressed by implementing an additional workflow to overwrite the distorted couch through a couch replacement. However, significant anatomical changes during treatment can be challenging for the DIR algorithm, especially when using the initial pCT as a reference for deformation. At times, the generated DD sCT may not accurately depict tissue changes, with discrepancies surpassing 1000 HU. This is evident during tumor shrinkage and bowel displacement. Consequently, DD sCT might yield results too similar to the original pCT, potentially overlooking dosimetric impacts and providing false negative results, which could delay necessary treatment plan adaptation. This lapse can arise from limitations of the DIR algorithm, mainly due to the constraints imposed by its regularization term. Conditions involving rapid anatomical changes necessitate utilizing an updated pCT as a reference or avoiding DD sCT to prevent false negative results.

HU discrepancies in IC sCT are primarily influenced by low-density regions. Users should be cautious of discrepancies in the water-equivalent thickness (WET) of materials in the proton beam path. When corrected HU values are applied, IC sCT provides dependable dosimetric results for H&N and brain regions. However, challenges arise in the thorax region. Both DD and IC sCTs have HU discrepancies, especially near the heart and esophagus boundary. Point dose metrics, such as D_0.1cc_ or D_max_, might not yield consistent comparisons to the vCT plan. IC sCT in the lung often mirrors CBCT quality, with widespread discrepancies attributed to the lung’s heterogeneous nature. The region’s inherent heterogeneous nature makes HU corrections challenging. Consequently, using IC synthetic CT alone in the thorax area requires caution due to potentially misleading dosimetric results.

### 4.2. Dosimetric Impact

This study focuses on evaluating the clinical viability of using CBCT-based sCTs as alternatives to vCT scans. The data show that all average deviations for dosimetric endpoints fall within 5%, and the majority of standard deviations lie within 10%. This suggests that the plans on sCTs closely align with those on vCTs, especially for non-moving target sites (e.g., brain and H&N), though notable image outliers warrant attention.

For treatment sites with moving targets, dosimetric discrepancies were found in both types of sCT. This accounts for the significant discrepancies observed in areas like the heart and esophagus with a small volumetric dose evaluator in the lung and breast group. The slow scanning speed of CBCT in combination with the moving anatomy could lead to a blurred boundary at the heterogeneous tissue interface. Consequently, this effect is propagated to CBCT-based synthetic CTs. Thus, leaning more towards volumetric or mean doses as more indicative metrics on sCTs is advised. In particular, the use of sCT is discouraged in the abdominal treatment region where the proton beam intersects with the bowels. In addition, the PCS algorithm was chosen for this study due to its computational efficiency and compatibility with the available TPS. However, it is crucial to acknowledge its inherent limitations, particularly when dealing with lung tissue. The PCS algorithm may not fully capture the intricacies of proton interactions within such heterogeneous tissues. Given the unique challenges posed by the combination of low-density and highly heterogeneous tissue in lung cases, future investigations employing the Monte Carlo calculation method may be warranted to gain a more profound understanding of the dosimetric impact.

The DD sCT struggles to capture the daily variances in bowel air/tissue content, often differing from CBCT representations. Conversely, IC sCT introduces significant artifacts derived from existing CBCT anomalies. For instances where the target volume is proximate and at the proton beam’s distal end, the bowel’s small volumetric dose evaluator could be misrepresented due to the sensitivity of proton ranges and suboptimal sCT bowel image quality. Bowel assessments should be confined to the three abdominal cases where the bowel is in direct contact with the treatment target and the proton beam’s distal edge targets bowel structures. The rigorous monitoring of bowel positioning and selective abdomen cases is suggested. For prostate patients with pelvic wall lymph nodes involved, the time between patient treatment and vCT varies the dosimetric impacts significantly due to the change in bladder volume between the vCT and CBCT. In our clinic, this interval averaged 30 min between patient treatment and vCT scan. However, these bladder volume discrepancies will become less relevant when sCT emerges as the primary tool, and vCT scans are phased out. Our current clinical protocol instituted pre-setup ultrasound bladder volume verification using an ultrasound bladder scanner and CBCT acquisition right before the commencement of the treatment. This protocol ensures that the sCT generated from CBCT accurately captures the patient’s anatomy for the treatment. The study highlighted noticeable fluctuations in minor bowel air movements in the abdomen and alterations in bladder volume at the prostate location when comparing CBCT to the reference vCT. Regarding the rectum, its elongated shape on the sCT exhibited displacement at the juncture where the CBCT FOV edge merges with the pCT. This peculiarity can considerably skew the volumetric evaluation of the rectum.

### 4.3. Summary

The precision of HU values and anatomy representation in sCT images is crucial for precise proton treatment planning to minimize the risk of false negatives or false positives during adaptive decision making. While sCTs have made significant strides in rectifying the HU values of CBCT to pCT standards, occasional factors like inadequate image deformation or HU adjustments can produce errant sCTs. The discrepancies, following identifiable patterns, can be systematically corrected. As a result, the quality of both DD sCT and IC sCT images remains comparable to conventional pCT for proton therapy. Table 1 summarizes the findings of each type of sCT studied, serving as a reference for users to select suitable cases for synthetic CT generation and be mindful of potential pitfalls.

While both commercialized information image systems offer automated workflows for sCT generation, users must carefully compare sCT quality with CBCT. Key areas of concern include metal artifacts, deformation accuracy, and geometric consistency with CBCT. Quality assurance tools that consider image and HU differences are necessary for future refinements, and standardized evaluation metrics for sCT quality are essential. Although most commercial systems automate the process, a thorough quality assessment, ideally visual, is advisable before using the sCT for plan evaluation. Techniques such as structure-based manual adjustments or overwriting discrepancies can enhance the results. Proper quality assurance and exclusion criteria should be in place to ensure the reliability of sCTs for clinical assessments.

## 5. Conclusions

This study outlines the advantages and potential challenges of two synthetic CT generation methods. By providing quantitative dose accuracy insights, it serves as a valuable guide for clinicians evaluating the results from these sCT methods. In the realm of PBS proton therapy, CBCT-based sCT can be clinically valuable both for monitoring treatment quality and initiating timely adaptive evaluations. Dosimetric concordance between sCT and vCT plans is typically greater in soft tissue regions compared to heterogeneous regions. In the early stage of clinical implementation, it is crucial to conduct a visual inspection of sCT image quality, along with accurate contour delineation and isocenter alignment propagated from the pCT to the sCT, as part of routine quality assurance. The judicious selection of appropriate volumetric constraints for sCT evaluation is advantageous in reducing the likelihood of false positive dosimetric results.

## Figures and Tables

**Figure 1 cancers-15-05101-f001:**
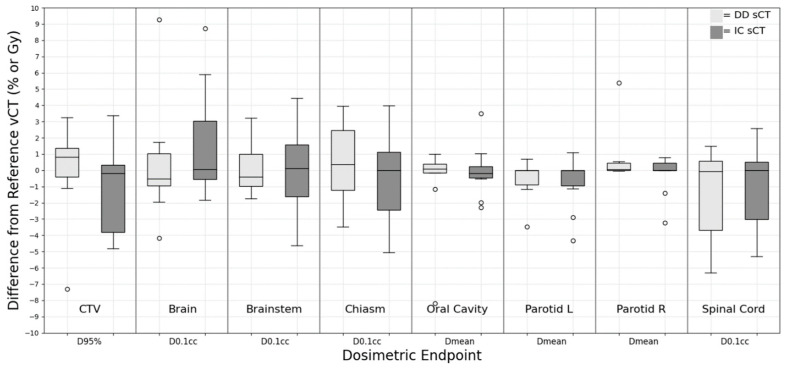
Dose difference between synthetic CT and reference verification CT on the same day for H&N treatment site.

**Figure 2 cancers-15-05101-f002:**
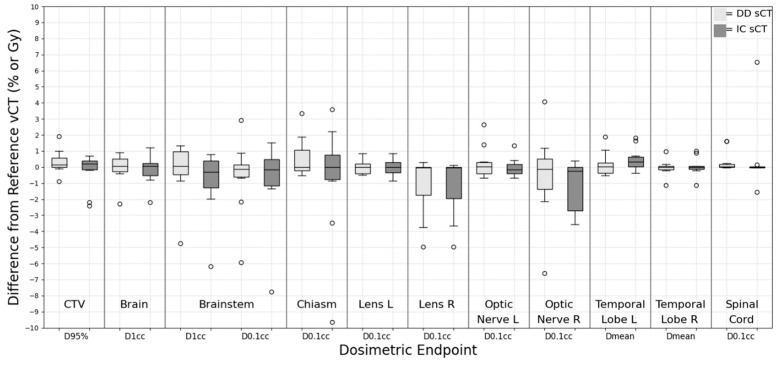
Dose difference between synthetic CT and reference verification CT on the same day for brain treatment site.

**Figure 3 cancers-15-05101-f003:**
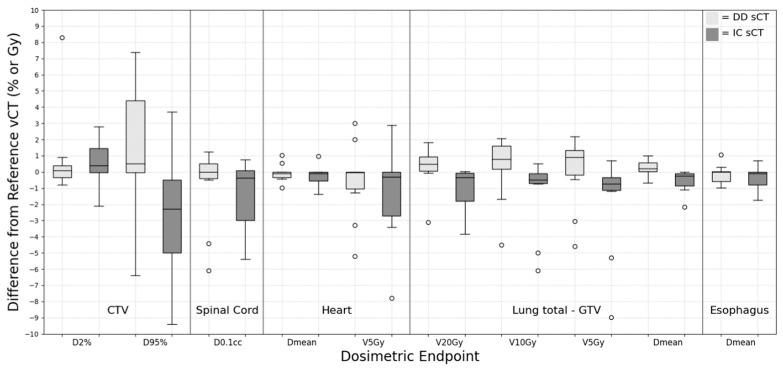
Dose difference between synthetic CT and reference verification CT on the same day for lung treatment site.

**Figure 4 cancers-15-05101-f004:**
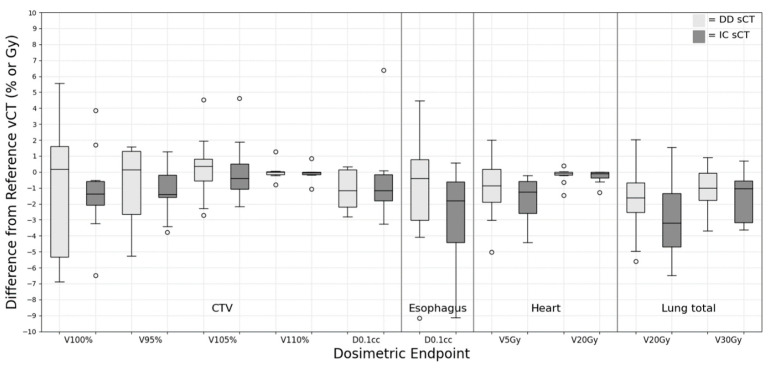
Dose difference between synthetic CT and reference verification CT on the same day for breast treatment site.

**Figure 5 cancers-15-05101-f005:**
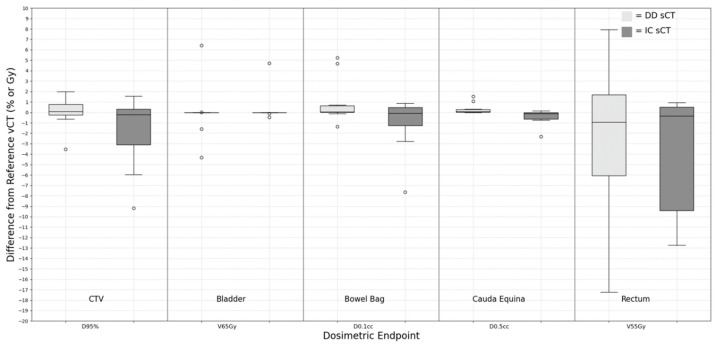
Dose difference between synthetic CT and reference verification CT on the same day for pelvis treatment site.

**Figure 6 cancers-15-05101-f006:**
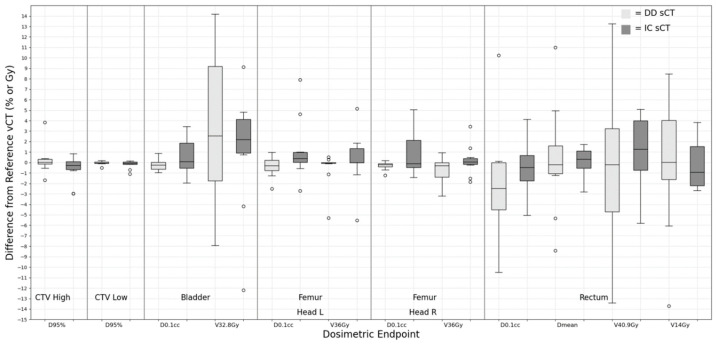
Dose difference between synthetic CT and reference verification CT on the same day for prostate with pelvic lymph node treatment site.

**Figure 7 cancers-15-05101-f007:**
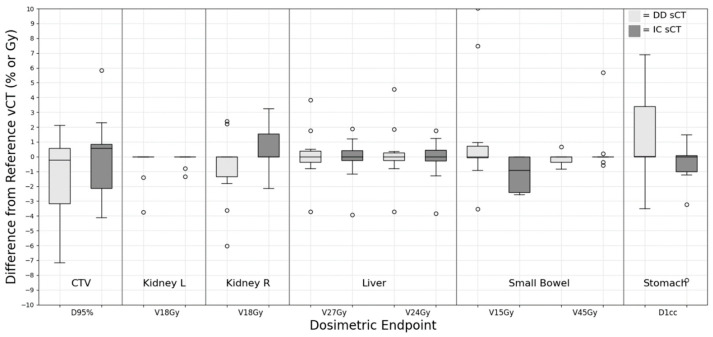
Dose difference between synthetic CT and reference verification CT on the same day for abdomen treatment site.

**Figure 8 cancers-15-05101-f008:**
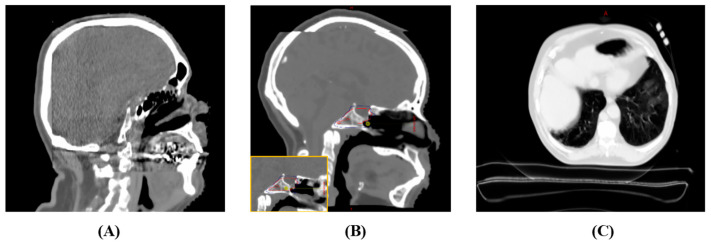
Depictions of discrepancies in synthetic CT images. (**A**) Case 1—An aliasing effect was observed near the posterior part of the brain where the CBCT FOV edge integrates with the planning CT. (**B**) Case 2—An aliasing effect was observed in the superior–inferior direction at the merged boundary, leading to an isocenter displacement error (isocenter in yellow circle) (for comparison, the original planning isocenter location is displayed in the bottom left). (**C**) Case 3—A distorted support couch structure in the DD sCT.

**Figure 9 cancers-15-05101-f009:**
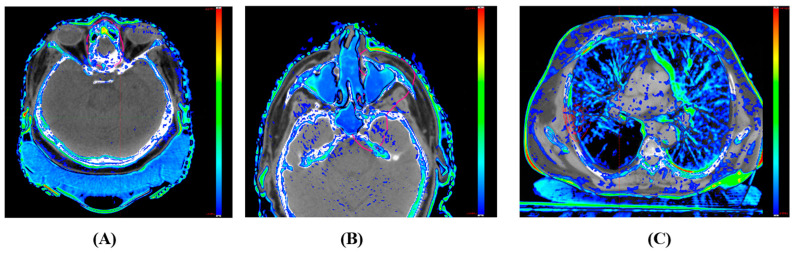
Differences in Hounsfield unit (HU) values between the image-corrected (IC) synthetic CT and the reference verification CT. The color bar indicates an HU discrepancy scale, spanning from 100 to 1000 HU. (**A**) Immobilization head cushion area; (**B**) nasal sinus region; (**C**) lung tissue and Vac-Lok ^TM^ region.

**Figure 10 cancers-15-05101-f010:**
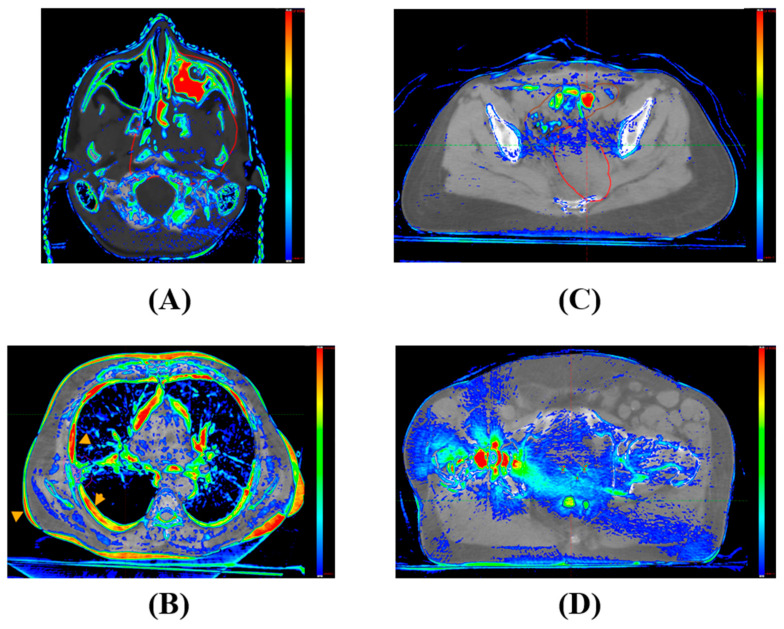
Disparities between the direct-deformable (DD) synthetic CT and the CBCT taken on the same day. Scenarios highlighted are: (**A**) tumor shrinkage within the cavity; (**B**) chest wall expansion; (**C**) discrepancies in bowel content; (**D**) hip prosthesis artifact. The color bar represents an HU difference scale from 100 to 1000 HU.

**Table 1 cancers-15-05101-t001:** Advantages and pitfalls of synthetic CT.

**Common Advantages of Synthetic CT**
Reduces the need for frequent patient verification CTs, especially for brain and breast cases in homogeneous tissue regions. Allows for plan evaluation based on the patient’s treated position and postures.
**Potential pitfalls of synthetic CT**
Direct-deformable (DD) synthetic CTs Degradation of image quality due to metal artifacts. Aliasing occurs near the edge of CBCT FOV merged with planning CT. Unnecessary deformations of rigid patient accessories (immobilization devices, treatment couch). Discrepancies between CBCT and synthetic CT near heterogeneous tissue interface (bone/air/tissue). Image-correction (IC) synthetic CTs Degradation of image quality due to metal artifacts Aliasing occurs near the edge of CBCT FOV merged with planning CT. Discrepancies in HU values were observed in low-density areas, such as head cushions, Vac-Lok ^TM^, and sinus cavities. Presence of diffused discrepancies in lung tissue.

## Data Availability

The data supporting the findings of this study are available within the article and from the corresponding author upon reasonable request. Due to privacy and ethical restrictions, the patient data and images cannot be publicly shared.

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
