# Peer review of "The Applications and Pitfalls of Cone-Beam Computed Tomography-Based Synthetic Computed Tomography for Adaptive Evaluation in Pencil-Beam Scanning Proton Therapy"

_cancers, 2023, doi:10.3390/cancers15205101_

Round 1
Reviewer 1 Report
The authors proposed a study investigating possible advantages and pitfalls of two syntethic CT generation methods (direct deformable (DD) and image correction (IC) methods) from CBCT for plan adaption in the realm of PBS proton therapy. Such analysis was carried out on a group of 70 patients, grouped for the most common type of tumors, comparing dosimetrical differences between nominal plan recalculation on re-valuative CT vs synthetic CTs with the two methods . Although the paper do not introduce any innovative aspect, the study is well structured and the analysis robust and well presented. I found it particularly useful from a clinical perspective and therefore I would suggest it for publication.
Major comments:
- Describe the plan adaptation protocol and how plan evaluation on synthetic CT form CBCT might affect/improve/modify it.
- Table 1 summarize the results from a theoretical point of view. I don’t find particularly interesting given that it is a resume of bibliographic results confirmed by the authors in this study. Instead, I would suggest the authors to introduce a new table, resuming the results of their dosimetrical analysis, grouped for anatomical district.
-
Minor comments:
Table 1. The table is not clear, please reshape it.
Line 460. Typo “occasional”
Reviewer 2 Report
This is a review of the manuscript entitled "The applications and pitfalls of CBCT-based synthetic CT for adaptive evaluation in pencil-beam scanning proton therapy," co-authored by P. Tsai et al.
This study intends to provide a systematic analysis/review of the utilization of CBCT-based synthetic CTs for adaptive RT in comparison to verification CT.
The work is overall well-described and scientifically rigorous. I think this is relevant to the community given the growing importance of adaptive RT.
I do not have significant concerns, but I do have a few comments/questions:
- 11 cases out of 81 - almost 15 %! - were marked as outliers for "unknown reasons". This is not negligible and not very satisfying of an analysis. Could the authors investigate and report on the reasons and potential fixes?
- I feel there would be more significance in presenting the absolute difference of the dose metrics rather than the difference. Of course, the standard deviation makes more sense as a +/- value.
- It is probably fair to discuss the limitations of the PCS algorithm. I am a strong advocate that Monte Carlo is not a necessity (unlike the general message nowadays); one needs a robust pencil beam algorithm - however, I don't believe PCS is peculiarly robust... I wonder how much this may impact the lung cases' results.
- Similarly, there is no information about beam arrangements. I understand that it is difficult for 70 cases, but it could explain some of the results. It would be interesting to at least investigate and comment.
- What is the rationale for the breast target metrics being so different from other sites? That does not make much sense to me.
Similarly, MHD and Lung V5 are relevant to breast cases.
- The authors show in Figure 8B a case with an isocenter shift due to imaging artifacts. I find this quite concerning. As briefly mentioned in the discussion, visual inspection would be beneficial and would have caught such an issue. A bit in contradiction with the authors, I actually don't think visual inspection is "advisable" but rather mandatory in these early days of clinical implementation.
- Figures 9 and 10 provide disparities between sCTs and vCT. Are these absolute disparities or two-sided? (I assume it is absolute, but would like confirmation).
It would also be of interest to show two images, one with error bars 0-100 HU (showing less significant dosimetric effects) and the current one 100-1000 HU, highlighting peculiarly significant discrepancies.
- I think Figure 11 is quite useless. It is self-evident that D0.1cc is less sensitive than Dmax… Dmax is used less and less, as it should.
- I was a bit surprised by some results for the prostate cases, but the discussion mentions that vCT was performed 30 minutes after patient treatment. This seems inappropriate. The vCT should be performed with the same delay post-dinking water as the pCT, and treatment...
- To me, it seemed all the DD results were superior to the IC results. Why wasn't this made more explicit in the discussion and summary?
Minor comments:
- Please stay consistent between each site; present the results in the same order (DD then IC, for instance) to improve readability.
- "Gy" should be "GyRBE"
- I find Table 1 is poorly formatted and hard to read.
Minor comments:
- line 129: "in this study" repeated.
- The beginning of each result section is repetitive "Figure X presents dose metrics...". Could be reworded to flow nicer in the overall picture.
- Line 310: spell out numbers up to ten.
- Line 411-412: "DIR algorithm" is repetitive.
- Line 436: "blurred"
